# Antifungal Activity of the Extract of a Macroalgae, *Gracilariopsis persica*, against Four Plant Pathogenic Fungi

**DOI:** 10.3390/plants10091781

**Published:** 2021-08-26

**Authors:** Latifeh Pourakbar, Sina Siavash Moghaddam, Hesham Ali El Enshasy, R. Z. Sayyed

**Affiliations:** 1Department of Biology, Faculty of Science, Urmia University, Urmia 5756151818, Iran; 2Department of Plant Production and Genetics, Faculty of Agriculture, Urmia University, Urmia 5756151818, Iran; sinamoghaddam2003@gmail.com; 3Institute of Bioproduct Development (IBD), Universiti Teknologi Malaysia (UTM), Skudai, Johor Bahru 81310, Johor, Malaysia; henshasy@ibd.utm.my; 4School of Chemical and Energy Engineering, Faculty of Engineering, Universiti Teknologi Malaysia (UTM), Skudai, Johor Bahru 81310, Johor, Malaysia; 5City of Scientific Research and Technology Applications (SRTA), New Burg Al Arab, Alexandria 21934, Egypt; 6Department of Microbiology, PSGVP Mandal’s, Arts, Science & Commerce College, Shahada 425409, India; sayyedrz@gmail.com

**Keywords:** biologically active compounds, fungistatic effects, quercetin, red alga, rosmarinic acid

## Abstract

Nowadays, the extract of seaweeds has drawn attention as a rich source of bioactive metabolites. Seaweeds are known for their biologically active compounds whose antibacterial and antifungal activities have been documented. This research aimed to study the profile of phenolic compounds using the HPLC method and determine biologically active compounds using the GC-MS method and the antifungal activity of *Gracilariopsis persica* against plant pathogenic fungi. *G. persica* was collected from its natural habitat in Suru of Bandar Abbas, Iran, dried, and extracted by methanol. The quantitative results on phenolic compounds using the HPLC method showed that the most abundant compounds in *G. persica* were rosmarinic acid (20.9 ± 0.41 mg/kg DW) and quercetin (11.21 ± 0.20 mg/kg DW), and the least abundant was cinnamic acid (1.4 ± 0.10 mg/kg DW). The GC-MS chromatography revealed 50 peaks in the methanolic extract of *G. persica*, implying 50 compounds. The most abundant components included cholest-5-en-3-ol (3 beta) (27.64%), palmitic acid (17.11%), heptadecane (7.71%), and palmitic acid methyl ester (6.66%). The antifungal activity of different concentrations of the extract was determined in vitro. The results as to the effect of the alga extract at the rates of 200, 400, 600, 800, and 1000 μL on the mycelial growth of four important plant pathogenic fungi, including *Botrytis cinerea, Aspergillus niger, Penicillium expansum,* and *Pyricularia oryzae*, revealed that the mycelial growth of all four fungi was lower at higher concentrations of the alga extract. However, the extract concentration of 1000 μL completely inhibited their mycelial growth. The antifungal activity of this alga may be related to the phenolic compounds, e.g., rosmarinic acid and quercetin, as well as compounds such as palmitic acid, oleic acid, and other components identified using the GC-MS method whose antifungal effects have already been confirmed.

## 1. Introduction

Algae are autotrophic organisms that contain chlorophyll but lack flowering organs or real roots, stems, and leaves and can convert solar energy into chemical energy by photosynthesis [1]. Macroalgae, or seaweeds, have long been used as a source of food, forage, fertilizer, and medication. Seaweeds provide many raw materials, including agar, algin, and carrageenan used in many industrial sectors. Moreover, they are consumed as food in many Asian countries [2] because they contain carotenoids, diet fibers, proteins, necessary fatty acids, vitamins, and minerals required for human nutrition [3]. Depending on their phylum, growth stage, and environmental conditions, algae can contain different amounts of bioactive compounds, including secondary metabolites, that could exhibit antiviral, antibacterial, and antifungal activities [4,5,6,7].

Plant diseases, especially those caused by plant pathogenic fungi, are key factors in the production and quality of crops [8,9,10] and in the postharvest life of fresh fruit and vegetables [11]. Among these pathogens, *Pyricularia oryzae* (*P. oryzae*) and *Botrytis cinerea* (*B. cinerea*) are two globally distributed fungi with a wide range of hosts, ranked as the first and second most important plant pathogens in the world based on their economic and scientific importance [12]. Known as the cause of rice blast disease, *P. oryzae* is the most damaging rice disease in the world. It is reported that it annually destroys 10–30% of the rice crop, sometimes reaching 100% in the case of an epidemic [12,13]. *B. cinerea* is responsible for various diseases, e.g., blossom blight, leaf spot, and fruit and bulb rot, both on farms and post-harvest crops. Since the organs of the infectious agent grow on the surface of the infected tissues, the developed infection is called gray mold [14]. *Aspergillus niger* is one of the most dominant species from the genus *Aspergillus*, which is responsible for black mold disease in fruits and vegetables [15,16]. It is also known as the common contaminant of foods, especially in sun-dried foods, grains, and nuts, and is the primary agent of postharvest rot of fruits and vegetables. In addition, it is regarded as an opportunistic pathogen of humans [17]. *Penicillium expansum* is known as the cause of blue mold or soft rot in many vegetables and fruits in the world. Its name originated from the blue-color conidium masses that it produces in the infected parts [18,19]. This species is the most important postharvest pathogen of apples and pears as it not only damages them extensively during storage but is also vital, as it produces the carcinogen toxin patulin in the infected fruits used to make fruit juice [20]. 

There are different ways to control diseases, e.g., the use of resistant cultivars, agronomic practices such as the use of healthy seeds, the balanced use of fertilizers, and biological and chemical control methods [21]. As a result of the frequent use of fungicides, some fungal isolates have become resistant to different fungicide groups [22]; therefore, these fungicides can be used limitedly in disease control [23,24]. It is, therefore, necessary to find new chemicals or use effective natural substances to control the disease.

Moisture on plants is a major factor involved in the fungal infections of these four fungal diseases. The current strategies to control fungal diseases include preventing moisture on leaf area for a prolonged period, the development of resistance in the host plants, and the application of fungicides. Nowadays, various artificial fungicides are applied for the protection of plants from these diseases [21]. Interest is growing in research on the industrial application of medicinal plants due to increasing attention to the use of natural antioxidants and fungicides as an alternative to synthetic ones, which are unsafe and toxic. Recently, many studies have reported the antifungal and antimicrobial activities of algae extracts, essential oils, and other materials. The number of compounds derived from various families of macroalgae, including green algae, brown algae, and red algae, is estimated at 40,000, which plays a key role in plant protection and improvement and is indeed a new approach for pest management [25,26]. 

*Gracilaria* is a genus of red algae with a global spread found in polar, moderate, and hot regions, but most of its species have been reported in tropical waters [27]. Research on the coastal areas in Southern Iran has detected a species of these algae in the Persian Gulf called *Gracilariopsis persica* (Rhodophyta) [28]. 

Most synthetic fungicides can potentially be harmful to the environment and can leave toxic residues in soil and/or crops [29]. Therefore, it is of crucial significance to find new biodegradable natural and eco-friendly bioactive compounds that possess potential biorational activity [30]. Accordingly, a goal we pursued in this research was to assess and examine natural active ingredients in the extract of *G. persica* using HPLC and GC-MS methods with a focus on its biorational effect (relatively non-toxic with few environmental side effects) against four pathogenic fungi.

## 2. Materials and Methods

### 2.1. Sampling from G. persica

Algae seedlings were collected from their natural habitat in Suru of Bandar Abbas, Iran at high water on November 16, 2018. In the habitat of *G. persica* on the transect, three transects were selected parallel to the coast. Samples were randomly taken by throwing 50 × 50 cm quadrats in five replications and cutting the seedlings in each quadrat with a spatula from their joining to the sand bed. The samples were placed within moist Kenaf fibers (damped by seawater) in multiple layers. They were then transferred to a laboratory with coolers [31]. The thalli collected from the natural habitat were placed in an aquarium for 24 h. The aquarium for the seawater was filled with 40-ppt saline water and kept at 25–27 °C with aeration. The thalli of the algae were rinsed with the water of the related aquarium to remove mud and unwanted epiphytes.

### 2.2. Extraction to Assess Antifungal Activity

The collected algae were dried for 1 week in the ambient shade drying conditions at an average temperature of 27 °C and after grinding, their powder was used for extraction. Therefore, 50 g of the ground alga’s dry matter was poured into a 500-milliliter Erlenmeyer flask where 200 mL of methanol was added. It was then shaken at 25 °C for 10 days, and its extract was filtered through a grade 2 Whatman^®^ filter paper. The extract was dried with an evaporator and kept in a sterilized sealed glass container in darkness at 4 °C [32]. Then, to check the antifungal effects, the dried extract was dissolved in water. 

### 2.3. Preparation of Methanolic Extract for HPLC and GC-MS Analysis

Two g of the powdered alga sample were added to 25 mL of methanol containing 1% acetic acid, placed in a magnetic shaker for 3 h, and filtered by grade 1 Whatman^®^ filter paper.

### 2.4. Analysis of Methanol Extract by Gas Chromatograph-Mass Spectrometry (GC-MS)

The methanolic extract of the alga was analyzed using Agilent 7890A Gas Chromatography/Mass Spectrometry (GC/MS) (Agilent 7890A, Agilent Technologies Inc., Santa Clara, CA, USA) and 5975 A mass spectrophotometer using an HP-5 MS capillary column (polydimethylsiloxane with a length of 30 m, an internal diameter of 0.25 mm, and a thickness of 0.25 μm). The initial temperature of the oven was set to increase from 80 to 180 °C at a rate of 8 °C/min. Helium was used as the carrier gas whose speed was 1 mm/min along the column length. The injection valve was set in the split mode at a ratio of 1:500 and an injection temperature of 250 °C. The mass spectrum was 40–500 mass/load and ionization energy of 70 eV. The whole run time was 55 min. The mass libraries Wily 2007 and NIST were employed to identify the compounds. Data were processed in the Windows-based Chemstation software. The relative percentage of the extract constituents was expressed in percent based on the peak level.

### 2.5. Measurement of Phenol Compounds by the HPLC Method

The phenolic compounds were isolated, detected, and quantified using High-Performance Liquid Chromatography HPLC (Agilent 1100, Agilent Technologies Inc., Santa Clara, CA, USA) equipped with a 20-microliter injection loop, a four-solvent gradient pump, a degasser, a column oven (set at 25 °C), and a diode array detector set at 250, 272, and 310 nm.

The isolation was carried out by an Octadecyl Saline Column (ODC) (with a length of 25 cm, an internal diameter of 4.6 mm, and a particle size of 5 μm, ZOR BAX Eclipse XDB, Germany). Data were processed using the Chemstation software. The mobile phase consisted of acetic acid (1%) (A) and acetonitrile (B) at a flow rate of 1 mL/min. The phenolic compounds were eluted under the following conditions: 1 mL/min of flow rate, the temperature of 25 °C, isocratic conditions from 0 to 10 min with 10% B, gradient conditions from 10 to 25% B in 5 min, from 25 to 65% B in 10 min, from 65 to 100% B in 15 min, followed by washing and reconditioning the column. The injection volume was 10 µL and phenolic compounds were detected at wavelengths of 250, 272, and 310 nm. The phenolic compounds were identified by comparing their relative retention times and compared with standards [33]. 

### 2.6. Preparation of Potato Dextrose Agar (PDA)

Fifty g of peeled Irish potatoes were rinsed after slicing and boiled for 1 h with distilled water in a conical flask. A muslin cloth was applied to filter the boiled potatoes and the volume was increased to 250 mL with water along with dextrose (5 g) and agar (3.5 g). The suspension was dissolved with heat and shaking, then left to cool down at room temperature for two min, then sterilized in an autoclave at 121 °C for one h.

### 2.7. In Vitro Antifungal Activity of the Extract

The antifungal effect of the extract of *G. persica* was checked using the following method. The fungi *B. cinerea*, *P. oryzae*, *A. niger*, and *P. expansum* were purchased from the Fungi and Bacteria Collection Center of Iran and cultured in a laboratory. To evaluate the antifungal activity of the alga extract, 200, 400, 600, 800, and 1000 μL of the extract were poured on sterile PDA plates. Distilled water was chosen as the control. Then, a ring with a diameter of 5 mm was taken from the margin of the actively growing colonies of the fungi and transferred to the center of the culture medium. All inoculated plates were kept in an incubator at 25 °C and were checked daily until the surface of the culture medium was entirely covered with the fungi in the control treatments. Then, the radial growth of the fungi was measured. The extract inhibitory effect on the mycelial growth of the fungi was calculated for each concentration by using the following formula [34]:(1)$$\mathrm{MGI}\;\left(\%\right)=\frac{dc-dt}{dc}\times 100$$
where MGI is the mycelial growth inhibition, *dc* is the mean diameter (mm) of the fungal mycelial growth in the control treatment, and *dt* is the mean diameter (mm) of the fungal mycelial growth extract-containing treatment.

### 2.8. Fungistatic or Fungicide Effect of the Extract

At the end of the trials, another experiment was conducted on the treatments in which no growth was observed to determine whether the extracts had killed the fungi (fungicidal effect) or had inhibited their growth temporarily (fungistatic effect). For this experiment, a PDA culture medium that contained no additive was prepared and poured into Petri dishes. The fungi-containing rings from the treatments that had no growth were transferred to the new culture medium. The Petri dishes were placed in an incubator at 25 °C and were re-checked after 7 days. The treatments in which the mycelial growth was observed after 7 days showed the fungistatic activity of the extract, while the lack of mycelial growth in the fungi-containing ring in the culture medium exhibited its fungicide activity.

### 2.9. Data Analysis

The data were subjected to the analysis of variance (ANOVA) and the comparison of means in the SPSS (Ver. 16) software package. The mean values were compared with Duncan’s multiple range test. In all graphs, the results were expressed in average values of three replications ± standard deviation (SD). Additionally, the graphs were drawn in MS-Excel (2016) software package.

## 3. Results

### 3.1. Results of GC-MS

The biologically active components in the methanolic extract of *G. persica* were detected using GC-MS with a running time of 55 min. The GC-MS chromatogram showed 50 peaks in methanolic extract, implying 50 compounds (Figure 1). The spectra of these compounds were compared with Wiley 7.0 and National Institute of Standards and Technology libraries. The detected compounds are listed in Table 1. The main fractions included cholest-5-en-3-ol (3. beta), palmitic acid, heptadecane, palmitic acid methyl ester, bicyclo [3.1.1] heptane, 2,6,6-trimethyl or pinane, isobutyl phthalate, *N*-cyano-*N*′, *N*′, *N*″, *N*″-tetramethyl-1,3,5-triazinetriamine, 5-thiazoleethanol, 4-methyl, and phytol.

### 3.2. Quantitative Analysis of Phenolic Compounds Determined Using HPLC

Figure 2 displays the standard HPLC chromatogram of all the recorded compounds at 250, 272, and 310 nm. According to Figure 2, all the studied compounds responded to the three spectra and were isolated successfully. The compounds of the alga extract were also separated using the same method at the same wavelengths, and they were identified using the standard graph (Figure 3a–c).

The quantitative results for the phenolic compounds using the HPLC method showed that the most abundant phenols in *G. Persica* were rosmarinic acid (20.9 ± 0.41 mg/kg DW) and quercetin (11.21 ± 0.20 mg/kg DW), and the least abundant was cinnamic acid (1.4 ± 0.10 mg/kg DW) (Figure 3 and Figure 4).

### 3.3. Antifungal Activity

Based on the results, as the concentration of the *G. persica* extract was increased, the growth of *B. cinerea*, *P. oryzae*, *A. niger*, and *P. expansum* was inhibited to a significantly higher extent (Figure 5). The lowest MGI in all four fungi was 35.18, 16.88, 24.50, and 37.21% recorded at the extract volume of 200 μL, respectively. The mycelial growth of *B. cinerea* and *P. expansum* was completely inhibited at an extract volume of 800 μL, and that of *P. oryzae* and *A. niger* was fully inhibited at the extract rate of 1000 μL (Figure 5).

The results as to the effect of the alga extract on the mycelial growth of the studied fungi revealed a clear dose–response of the extract. The 100% inhibition was achieved at 800 and 1000 μL for *B. cinerea* and *P. expansum* (Appendix A) and 1000 μL for *P. oryzae* and *A. niger* (Appendix A).

#### Fungistatic

The fungistatic effect of various concentrations of the extract was observed as inhibition of mycelial growth. However, re-culturing the mycelial growth following the re-inoculation into the fresh medium resulted in the re-growth of the fungal mycelium in the new medium, implying that the alga extract could only inhibit the growth of the fungi (fungistatic).

In the study of fungicidal or fungistatic effects of extracts at concentrations that completely inhibited mycelium growth, the re-cultures resulting from the transfer of the fungi-containing ring to the PDA culture medium showed that the fungi reappeared and grew after 5–7 days on the fresh culture medium. The results revealed that *P. oryzae* and *A. niger* started to grow after 5 days and *P. expanstum* and *B. cinerea* started to grow after 7 days. During the study period (7 days), the growth of mycelium in the re-cultures of the control fungi was completely evident from the second day, and it was much higher (up to 90%) at the end of 7 days than the re-cultures performed at concentrations in which mycelium growth was completely inhibited. These results indicate that the algae extract only inhibited the growth of fungi and fungicidal properties were observed in none of the extract concentrations.

## 4. Discussion

The results obtained from the GC-MS analysis of this study specify that *G. persica* showed a large number of bioactive compounds with antioxidant, antibacterial, and antifungal properties. There are various reports about the compounds derived from macroalgae as a potential source of biochemical and medicinal properties including antibacterial [57], antifungal [58], antiviral [59], antioxidant [60], and anti-inflammation activities [61].

The results of the present study revealed that the extract of *G. persica* had a high potential to inhibit the mycelial growth of plant pathogenic fungi. Previous research findings have indicated that alga extracts have agents for the biological control of the growth of hyphae and germination, an increase in intracellular holes (vacuolization), and the disruption of the functioning of fungal cells [62]. On the other hand, research on different species of *Gracilaria* has revealed this genus’ antibacterial and antifungal activities. In this regard, Singh and Raadha [63] studied the *G. corticata* extract. They found that this species, at a rate of 1000 μL, could inhibit the growth of human pathogenic bacteria and fungi, including *Salmonella typhimurium*, *Escherichia coli*, *Staphylococcus aureus*, and *Candida albicans*, and was a natural source of antibiotics. Dayuti [64] also reported the antibacterial activity of *G. verrucosa* against *S. typhimurium* and *E. coli*. A study on the antifungal and antibacterial properties of the *G. confervoides* extract showed that 100 μL of the *G. confervoid* extract could prevent the aerial mycelial growth of the cucumber pathogens *Rhizoctonia solani* and *Macrophomina phaseolinae* [58]. This is consistent with our findings as to the fungistatic activity of the *G. persica* extract. In an experiment, Kolanjinathan and Stella [65] found that *G. corticata* could inhibit the growth of *Aspergillus flavus*, *A. fumigatus*, and *A. niger*, as well as the human pathogen *albicans*. The minimum inhibitory rate of this extract was estimated at 2–16 mg/mL. This is in agreement with our findings as to the antifungal property of *G. persica* in inhibiting the growth of *A. niger*.

The results of HPLC analysis in our study showed that *G. persica* is rich in polyphenolic compounds. It is now well established that the antifungal activity of alga extracts may be related to the presence of phytochemicals, e.g., tannins and phenols. Phenolic compounds are likely to influence the growth and metabolism of fungi [66]. Sea resources are the most enormous remaining reservoirs of natural molecules, which are assessed for therapeutic activities and provide valuable ideas for developing new medications against cancer, microbial infections, and inflammations [67]. Although terrestrial biodiversity constitutes the basis of the pharmaceutical industry, oceans have rich biodiversity and can produce commercially invaluable modern compounds. A comparison of our findings with those of other studies confirms that *G. persica* is a rich source of biological components.

Some phytocompounds, which combine alkaloids, flavonoids, and saturated and unsaturated fatty acids, have antimicrobial, anti-inflammation, anti-cancer, anti-coagulant, and anti-arrhythmic activities. According to the results of GC-MS, most of the compounds detected in the methanolic extract of *G. persica* (36 out of the 50 detected components) are compounds whose antimicrobial and antifungal activities have already been documented. These interpretations corroborate the findings of current research (Table 1).

The results of HPLC revealed that *G. persica* had invaluable phenolic compounds, e.g., rosmarinic acid and quercetin, implying its antioxidant effect. It has been reported that the difference in plant extracts’ biological activity depends on their constituents; therefore, a single compound may be responsible for an extract’s effects alone or in synergy with other compounds [68].

Polyphenols are a remarkable group of plant metabolites that have an efficient antimicrobial performance. A number of studies have postulated about the interaction of the synergy of polyphenols with antibiotics against microbial resistance, e.g., epigallocatechin gallate of green tea [69], tellimagrandin I, and rugosin B of *Rosa floribunda* ‘Dubline Bay’ (aka ‘MACdub’) [70], and the synergy of rosmarinic acid and antibiotics against methicillin-resistant *Staphylococcus aureus* [71]. Likewise, the antimicrobial effect of quercetin as the second dominant phenolic compound in *G. persica* against *Escherichia coli*, *Staphylococcus aureus*, and *Pseudomonas fluorescens* [72], and the antifungal effects of quercetin and rutin against *Cryptococcus* spp. [73] have been documented. It has been reported that these compounds can change the structure of the cell membrane, thereby destroying the plasma membrane integrity in fungal cells and increasing its permeability, which results in an increase in the K^+^ outflow from the cytoplasm of fungal cells. These effects may cause the polarity of the membrane to be lost by altering ion transport, or they may reduce energy production (ATP) by altering the membrane structure through impairing glucose uptake or inhibiting the enzymes involved in oxidative stress or phosphorylation precursor. The increase in cytoplasm membrane permeability ultimately leads to cell death due to the dispersion and loss of the cell pH gradient, a decline in the ATP level, and the failure of the proton driving force. Indeed, nutrient uptake, nucleic acid synthesis, and ATP*ase* activity sections are most damaged in the tissues of the fungi [74,75].

The experimental evidence of current research regarding the examination of polyphenolic compounds in *G. persica* showed that quercetin was the highest among the flavonoid compounds. The possible activity of flavonoids may also be involved in mitochondrial damage and ROS production by inducing the transcription factors related to apoptosis and increasing the level of proptose proteins [76]. Hwang et al. [77] revealed that flavonoids disturbed the performance of mitochondria in the *C. albicans* strain by increasing the ROS level. The elevated intracellular ROS level and the disturbed mitochondrial performance play a significant role in apoptosis induction [78,79]. Quercetin has two aromatic rings in its structure and can penetrate the phospholipid membrane [80], where it damages DNA by inducing oxidative damage and, finally, cause cell death by apoptosis, which is an irreversible process [81].

## 5. Conclusions

Herbal medicines play a significant role in human health and are an inspiring source of new medicinal compounds. It can be concluded from the results of the present study that *G. persica* has a high potential to be used in the pharmaceutical industry to improve health owing to its different compounds with antimicrobial activity.

The inhibitory effect of the alga on the growth of plant pathogenic fungi *in vitro* is apparent from our experimental evidence. Thus, this alga might be used to produce an environmentally friendly, reliable, and economic antifungal agent to control *B. cinerea*, *P. oryzae*, *A. niger*, and *P. expansum*. This can be a high potential alternative to highly toxic chemical fungicides in plant disease management. The fungicidal efficacy still has to be shown in vivo. Above all, our results suggest that the algae in the Persian Gulf and Oman Sea can be a rich source of different macroalgae species with unique antimicrobial activities, which may have various applications in agriculture and the control of plant disease in the future.

## Figures and Tables

**Figure 1 plants-10-01781-f001:**
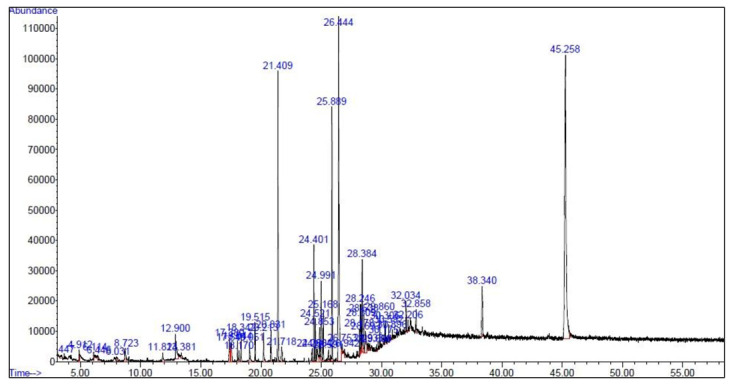
Gas chromatography and mass spectroscopy chromatogram of the methanolic extract of the *Gracilariopsis persica* alga.

**Figure 2 plants-10-01781-f002:**
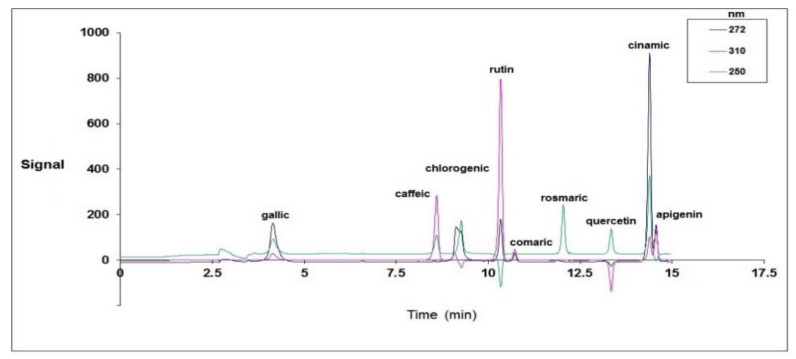
The standard graph of the phenolic compounds in the HPLC chromatogram.

**Figure 3 plants-10-01781-f003:**
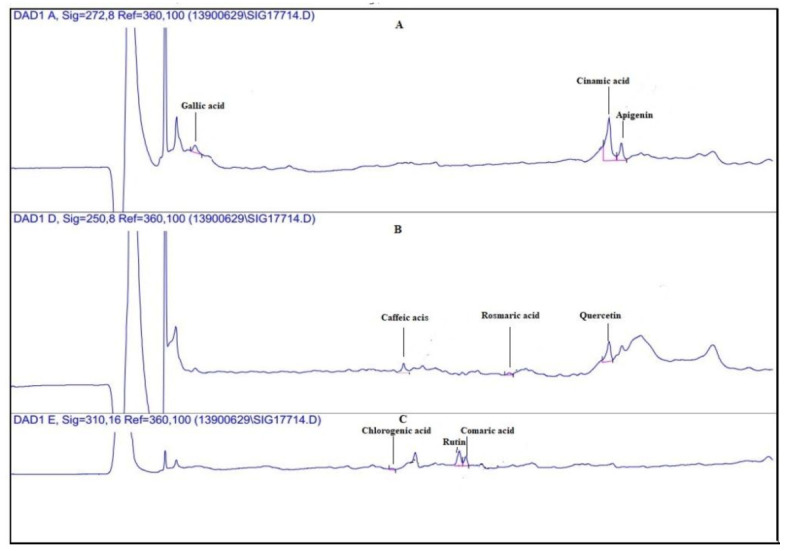
The diagram of identifying the phenolic compounds using the HPLC method at three wavelengths of (**A**) 272 nm, (**B**) 250 nm, and (**C**) 310 nm.

**Figure 4 plants-10-01781-f004:**
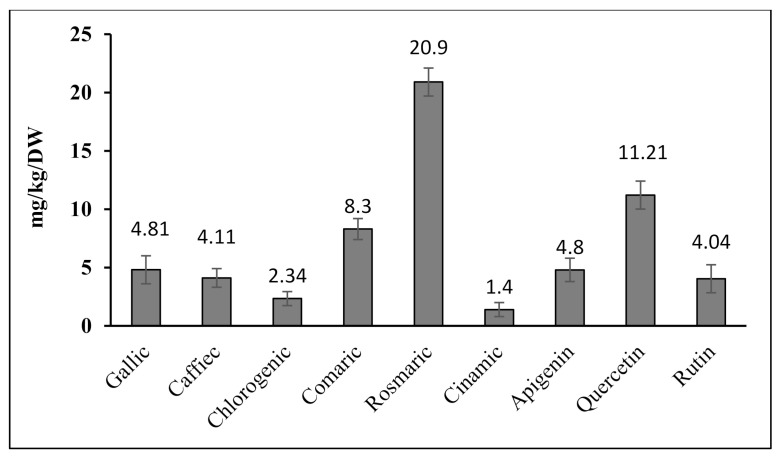
The quantity of the phenolic compounds identified using the HPLC method in *G. persica* (columns are mean SD ± 3 replications and vertical bars show standard deviation).

**Figure 5 plants-10-01781-f005:**
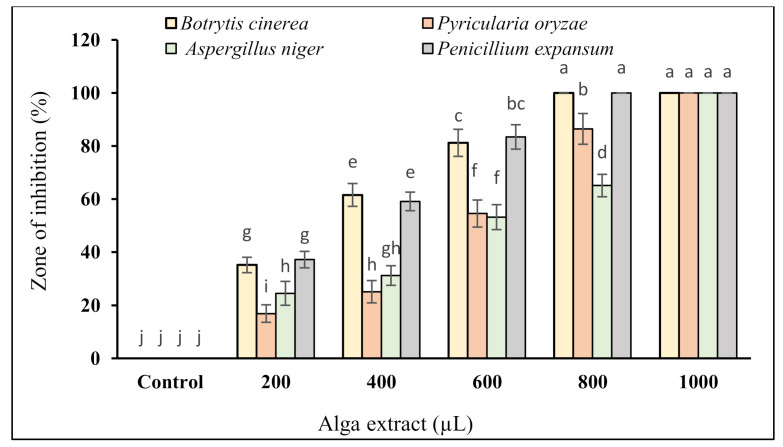
The effect of the *G. persica* extracts at volumes, 200, 400, 600, 800, and 1000 μL on the mycelial growth inhibition of *B. cinerea*, *P. oryzae*, *A. niger*, and *P. expansum* (columns are mean SD ± 3 replications, vertical bars show standard deviation, and different letters imply significant differences at the *p* < 0.05 level based on Duncan’s test.).

**Table 1 plants-10-01781-t001:** The phytocomponents were detected in the methanolic extract of the *Gracilariopsis persica* using gas chromatography and mass spectroscopy.

Retension Time	Name of the Compound	Area %	Bioactivity	Reference
3.447	Boron, trihydro (N-methylmethanamine-, (T-4)-	0.36		
4.912	2 (3H)-Furanone, dihydro-4-hydroxy-	0.45	Antibacterial	[35]
6.114	1-Hepten-3-ol	0.21	-	
6.646	D-Ribonic acid, 2,3-O-(ethoxymethylene)-	0.37	-	
8.031	2-Octyldodecan-1-ol	0.19	-	
8.723	Docosane	0.88	Antifangal	[36]
11.824	2-Decyne	0.46	-	
12.900	5-Thiazoleethanol, 4-methyl-	2.12	Antifungal, anti-inflammatory, anti-allergic	[37]
13.381	Carvacrol	0.25	Anti-inflammatory, antioxidant, antitumor, antibacterial	[38]
17.380	2-(1,4,4-Trimethyl-cyclohex-2-enyl)-ethanol	1.21		
17.541	Decamethylpentasiloxane	0.83	Antibacterial, antifungal	[39]
18.044	2-cyclohexene-1-one	0.62	Antibacterial	[40]
18.170	10-Methyl-9-oxabicyc lo[6.4.0]dodecan-1(8)-ene	0.43	-	
18.342	2-(4H)-Benzofuranone, 5,6,7,7a-tetrahydro-4,4,7a-trimethyl-	0.76	Antimicrobial preservative, antifungal, antibacterial	[41]
19.051	2-Propenoic acid, 3-(1-aziridinyl)-,methyl ester	0.85		
19515	Diethyl Phthalate	1.06	Antioxidant	[42]
20.213	Methanone, diphenyl-	0.98	Antibacterial, antifungal	[43]
20.831	Cyclododecasiloxane, tetracosamethyl-	1.37	Antimicobial, antirheumatic antispasmodic	[44]
21.409	Heptadecane	7.71	Anticancer, anti-inflammatory	[38]
21.718	n-Dodecanal	0.66	Antibacterial	[41]
24.230	Cyclododecasiloxane, Tetracosamethyl-	0.39	Antimicobial, antirheumatic antispasmodic	[44]
24.662	Caffeine	0.97	Antibacterial, antifungal	[45]
24.401	Bicyclo[3.1.1] heptane, 2,6,6-trimethyl Pinane	3.17	Antifungal	[46]
24.521	11-Dodecen-2-one	1.13		
24.853	E-10-Methyl-11-tetradecen-1-ol acetate	0.76		
24.991	Isobutyl phthalate	2.25	Antibacterial, antifungal	[42]
25.168	Neophytadine	1.66	Antimicrobial, anti-inflammatory	[47]
25.591	2-Propenoic acid, 2-methyl-	0.43	Antibacterial, antifungal	[48]
25.889	Palmitic acid methyl ester	6.66	Antibacterial, antifungal	[41]
26.444	Palmitic acid	17.11	Antibacterial, antifungal	[49]
26.753	Oleic acid	0.24	Antibacterial, antifungal	[49]
26.942	1-Decanol, 2-hexyl-	0.46	Antimicrobial	[50]
28.178	Linolelaidic acid, methyl ester	0.66	Antibacterial, antifungal	[41]
28.246	Oleic acid methyl ester	1.18	Antibacterial, antifungal	[51]
28.309	Oleic acid methyl ester	0.94	Antibacterial, antifungal	[41]
28.384	Phytol	2.34	Antibacterial, antifungal	[52]
28.538	Stearic acid methyl ester	1.01	Antibacterial, antifungal	[41]
28.693	9-Hexadecenoic acid	1.73	Antibacterial, inflammator	[52]
28.933	9-Hexadecenoic acid	0.32	Antibacterial, inflammator	[52]
29.608	E-11-Tetradecenoic acid	0.25	Antibacterial, antifungal	[52]
29.860	Hexadecanedioic acid	0.93	Antimicrobial, anti-inflammatory	[52]
30.175	9-Borabicyclo[3.3.1]nonane, dimethylamino)propyl]	0.27	-	
0.306	(-)-18-noramborx	0.45	-	
30.592	Trisiloxane, 1,1,1,3,5,5,5-heptamethyl	0.53	Antifungal	[53]
30.896	Hexamethylcyclotrisiloxane	0.28	Antimicrobial	[54]
32.034	1,3-Bis(trimethylsilyl)benzene	0.79	-	
32.206	p-Bis(trimethylsilyl)benzene	0.47	-	
32.858	Phthalic acid,bis(2-ethylhexyl) ester	0.75	Antibacterial, antifungal	[35]
38.340	N-Methyl-1-adamantane acetamide	2.52	Antimicrobial	[55]
45.258	Cholest-5-en-3-ol (3.beta.)-	27.64	Antibacterial	[56,57]

## Data Availability

Not applicable.

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
