# Peer review of "Antifungal Activity of the Extract of a Macroalgae, Gracilariopsis persica, against Four Plant Pathogenic Fungi"

_plants, 2021, doi:10.3390/plants10091781_

Round 1

Reviewer 1 Report

In the present study, the Authors examined the antifungal activity of the extract of the red alga Glacilariopsis persica. The study is relevant and actual, the constructed experiments and the measurements carried out are appropriate. However, all parts of the manuscript requires more or less careful revision to create a coherent article. It is essential to check the English of the manuscript, because it is full of incorrect grammar (wrongly used prepositions, terms, strange sentence structures, etc.). In its present state it is difficult to understand what Authors want to say in many cases.

Abstract

P1L31: The correct name of the fungus is Penicillium expansum.

Introduction

P1L40: Please use „autotrophic organisms” instead of „autotroph primitives”.

P2L57: Please rephrase: “sometimes amounting to 100% of the crop in case of the epidemic” to “sometimes reaching 100% in case of epidemic”

P2L58: Citations should be numbered consecutively (so [8,7] is not correct).

P2L76: This sentence looks like not to be finished.

P2L81-83: This sentence is not completely clear… Does this means that the amount of compounds derived from macroalgae is estimated 40,000?

P2L78-96: This part of the Introduction is a little bit messy. It should briefly present the possibilities of control of fungal diseases, including the disadvantages of synthetic fungicides, the possible advantages of fungicides with biological origins, the current state of the latter in practice. Study questions or hypotheses should be worded, which could lead the attention of the readers and also helps the authors themselves, what and how they would like to present.

Materials and methods

P3L110: What does it mean “dried in the shade”? In complete darkness, or in different (shady) states of light? What authors think, does drying on room temperature have any effects on the composition of the algal material (e.g. because of ongoing enzymatic reactions)?

P3L113: The extract was filtered, not infiltrated (these two words mean different things). Please use the correct word throughout the manuscript.

P3L121: The type of the instrument should be deleted before the parenthesis (chromatography is the method, not the instrument, the type of the chromatograph is given within the parentheses).

P3L139: Octadecyl saline column, I guess.

P3L141: I think, Authors wanted to say: “Data were processed using Chemstation software.”

P3L141-145: The sentence about the elution program need to be clarified.

P4L147-158: It is not clear, what the control was. Cultures without any treatment, I guess. It should be given in this section. How did the authors exclude the possible negative effects of the diluent (methanol)? Were there cultures treated only with methanol?

P4L148: of G. persica

P4L151-154: Not all of the readers are familiar with PDA plates, so it should be given that the specified extract volumes were given to how much culturing medium. To give the % concentration (v/v) of the individual treatments would be the best.

P4L170-172: This sentence should be rephrased: in my opinion “mycelial growth was not observed” means the same as “lack of mycelial growth”.

P4L175-176: “The mean values were compared with Duncan’s multiple range test.”

Results

P5L191: Title of Table 1: Why cannot the term be simply “components”? We can be sure, that the components are not “zoocomponents”, since they are from a red alga (by the way, the word “phycocomponents” would be the most precise, since algae are not plants according to the present state of science…).

P6L193: Use the term “phenolic compounds”. Similarly at other parts of the manuscript.

Figure 3: Authors should “zoom” into the chromatograms, because the peaks seem smaller than noise in these pictures. I.e. it is not convincing at all, that the compounds are really there.

Figure 5: The name of Penicillium expansum is wrong on the figure.

Figure 6-7: In my opinion, these figures should be presented as supplementary materials. Some of the pictures seem to be distorted – I assume, that during their processing, the original height-to-width ratios were changed, which is unfortunate in the case of biological material, where the real shape could be very informative (and sometimes absolutely necessary).

P10L236-241: I think, this paragraph should be part of 3.3., it is not necessary at all to put it in a separated sub-section.

Discussion

The Discussion section is not well constructed. There is information about everything required to explain the results, but somehow the text is not coherent as a whole. It definitely should be improved.

P10L243: If an organism is considered to be a plant, it is definitely eukaryotic, since there are no prokaryotic plants. Eukaryotic algae are not plants according to the present state of science, they are protists (even the ones with root-, stalk- and leaf-like structures).

P10L247-248: The cited Authors and publication year should be deleted.

P10L257: Candida albicans is not a bacterium.

P11L288-: There are a lot of literature in this part, which sounds more like an Introduction than a Discussion. I guess, Authors try to explain in this part, why is it an important result that rosmarinic acid is the main compound of the studied red alga. Somehow all the important information should be summarized in context, not just sentences after each other. The long description of the mode of action of phenolic compounds is not necessary, since the Authors did not study the mode of action.

P11L307-308: This sentence seems unfinished

Reviewer 1 Report

Comments and Suggestions for Authors

  • In the present study, the Authors examined the antifungal activity of the extract of the red alga Glacilariopsis persica. The study is relevant and actual, the constructed experiments and the measurements carried out are appropriate. However, all parts of the manuscript requires more or less careful revision to create a coherent article. It is essential to check the English of the manuscript, because it is full of incorrect grammar (wrongly used prepositions, terms, strange sentence structures, etc.). In its present state it is difficult to understand what Authors want to say in many cases.

Authors’ response: The MSS is thoroughly revised

Abstract

  • P1L31: The correct name of the fungus is Penicillium expansum .

Authors’ response: Corrected  (P1L35)

 Introduction

  • P1L40: Please use „autotrophic organisms” instead of „autotroph primitives”.

Authors’ response: Corrected (P2L45).

.

  • P2L57: Please rephrase: “sometimes amounting to 100% of the crop in case of the epidemic” to “sometimes reaching 100% in case of epidemic”

Authors’ response: Corrected (P2L65)

  • P2L58: Citations should be numbered consecutively (so [8,7] is not correct).

Authors’ response:  Corrected (P2L66).

  • P2L76: This sentence looks like not to be finished.

Authors’ response: Corrected (P2L87-88).

  • P2L81-83: This sentence is not completely clear… Does this means that the amount of compounds derived from macroalgae is estimated 40,000?

Authors’ response: This is exactly what it means and the text has been modified (P3L98-101).

  • P2L78-96: This part of the Introduction is a little bit messy. It should briefly present the possibilities of control of fungal diseases, including the disadvantages of synthetic fungicides, the possible advantages of fungicides with biological origins, the current state of the latter in practice. Study questions or hypotheses should be worded, which could lead the attention of the readers and also helps the authors themselves, what and how they would like to present.

Authors’ response: Corrected (P2&P3L89-117)

Materials and methods

  • P3L110: What does it mean “dried in the shade”? In complete darkness, or in different (shady) states of light? What authors think, does drying on room temperature have any effects on the composition of the algal material (e.g. because of ongoing enzymatic reactions)?

Authors’ response: The meaning of the sentence is the same as light shade, not darkness. Heat usually changes some of the plant active ingredients or emits volatile compounds from the plant. Therefore, the best way to keep these compounds is to dry them in the shade (PP3L133).

  • P3L113: The extract was filtered, not infiltrated (these two words mean different things). Please use the correct word throughout the manuscript.

Authors’ response: Corrected (P3L137).

  • P3L121: The type of the instrument should be deleted before the parenthesis (chromatography is the method, not the instrument, the type of the chromatograph is given within the parentheses).

Authors’ response: Corrected (P4L147).

  • P3L139: Octadecyl saline column, I guess.

Authors’ response: Corrected (P4L167)

  • P3L141: I think, Authors wanted to say: “Data were processed using Chemstation software.”

Authors’ response: Corrected (P4L169-170).

  • P3L141-145: The sentence about the elution program need to be clarified.

Authors’ response: Corrected (P4L170-179).

  • P4L147-158: It is not clear, what the control was. Cultures without any treatment, I guess. It should be given in this section. How did the authors exclude the possible negative effects of the diluent (methanol)? Were there cultures treated only with methanol?

Authors’ response: After methanolic extraction, the extract was dried by evaporator and then dissolved in water to measure antifungal activity. In the control, instead of the extract, the same amount of distilled water was added to the culture medium. These items were added to the materials and methods section (P3L139-140).

  • P4L148: of G. persica :

Authors’ response: Corrected (P5L188).

  • P4L151-154: Not all of the readers are familiar with PDA plates, so it should be given that the specified extract volumes were given to how much culturing medium. To give the % concentration (v/v) of the individual treatments would be the best.

Authors’ response: Preparation of Potato Dextrose Agar (PDA) added in Materials and Methods section (P4L180-186).

  • P4L170-172: This sentence should be rephrased: in my opinion “mycelial growth was not observed” means the same as “lack of mycelial growth”.

Authors’ response: Corrected (P5L207-208 & L215-216).

  • P4L175-176: “The mean values were compared with Duncan’s multiple range test.”

Authors’ response: Corrected (P5L219-221).

Results

  • P5L191: Title of Table 1: Why cannot the term be simply “components”? We can be sure, that the components are not “zoocomponents”, since they are from a red alga (by the way, the word “phycocomponents” would be the most precise, since algae are not plants according to the present state of science…).

Authors’ response: Corrected (P6L239).

  • P6L193: Use the term “phenolic compounds”. Similarly at other parts of the manuscript.

Authors’ response: Corrected (P7L241& P8L247&P9L252).

  • Figure 3: Authors should “zoom” into the chromatograms, because the peaks seem smaller than noise in these pictures. I.e. it is not convincing at all, that the compounds are really there.

Authors’ response: Corrected (P8LFig. 3).

  • Figure 5: The name of Penicillium expansum is wrong on the figure.

Authors’ response: Corrected (Supplementary).

  • Figure 6-7: In my opinion, these figures should be presented as supplementary materials. Some of the pictures seem to be distorted – I assume, that during their processing, the original height-to-width ratios were changed, which is unfortunate in the case of biological material, where the real shape could be very informative (and sometimes absolutely necessary).

Authors’ response:Figures were transferred to supplementary in the original height to width..

  • P10L236-241: I think, this paragraph should be part of 3.3., it is not necessary at all to put it in a separated sub-section.

Authors’ response: sub-section removed (P10L286).

 Discussion

  • The Discussion section is not well constructed. There is information about everything required to explain the results, but somehow the text is not coherent as a whole. It definitely should be improved.

Authors’ response: Improved (P11&12 L307-394).

  • P10L243: If an organism is considered to be a plant, it is definitely eukaryotic, since there are no prokaryotic plants. Eukaryotic algae are not plants according to the present state of science, they are protists (even the ones with root-, stalk- and leaf-like structures).

Authors’ response: Corrected (P11L308).

  • P10L247-248: The cited Authors and publication year should be deleted.

Authors’ response: Corrected (P11L313-314).

  • P10L257: Candida albicans is not a bacterium.

Authors’ response: Corrected (P11L323).

  • P11L288-: There are a lot of literature in this part, which sounds more like an Introduction than a Discussion. I guess, Authors try to explain in this part, why is it an important result that rosmarinic acid is the main compound of the studied red alga. Somehow all the important information should be summarized in context, not just sentences after each other. The long description of the mode of action of phenolic compounds is not necessary, since the Authors did not study the mode of action.

Authors’ response: Corrected (P12L371).

  • P11L307-308: This sentence seems unfinished

Authors’ response: Corrected (P12L375-379).

Reviewer 2 Report

Title: I suggest  ‘Antifungal activity of the extract of a macroalga, Gracilariopsis persica, against four plant pathogenic fungi’ 

Line 42: ‘have long been used to source food,..’ change in ‘have long been used as source of..’ 

Line 44’.. and carrageenan, for industrial products.’ Change in ‘ .. and carrageenan used in many industrial sectors.’ 

Vedere ‘contain in linea 45 e l’have riga 47 

Line 46-48: Owing to their diversity, algae have various compounds in different growth stages and are regarded as 47 rich sources of compounds generally known as secondary metabolites. These secondary metabolites can display diversity of their secondary metabolites, seaweeds are a potential source of antimicrobial 49 materials and exhibit antiviral, antibacterial, and antifungal activities [5]. 

I suggest: Depending to their phylum, growth stage and environmental conditions, algae can contain different amount of bioactive compounds, including secondary metabolites that could exhibit antiviral, antibacterial, and antifungal activities (references). 

Line 50: the authors mentioned a list of activity displayed by algae, but the reference is about only antibacterial activity. I suggest some articles for the other ones to be included: 

Righini, H., Roberti, R. & Baraldi, E. Use of algae in strawberry management. J Appl Phycol 30, 3551–3564 (2018).   

Angélica Ribeiro Soares et al. 2012. Antiviral activity of extracts from Brazilian seaweeds against herpes simplex virus. Rev. Bras. Farmacogn. Braz. J. Pharmacogn. 

Line 53: ‘are two globally distributed fungi with’ change in ‘..are two globally distributed plant pathogenic fungi with..’ 

Line 51-55: My suggestion: Plant diseases, especially those caused by plant pathogenic fungi, are key factors in the production and quality of agricultural crops and in postharvest life of fresh fruit and vegetables. Among these pathogens, Pyricularia oryzae (P. oryzae) and Botrytis cinerea (B. cinerea) are two globally distributed fungi with a wide range of hosts, ranked as the first and second most important plant pathogen in the world based on their economic and scientific importance [7]. 

Line 55-59: Please rewrite. 

Lines 71-86: Please, rewrite. It is confusing. The authors have mixed many concepts. Follow a logical thread, starting explaining as disease control is usually based on, the problems risen from the use of synthetic products 

Reviewer 2 Report

Comments and Suggestions for Authors

  • The first part on chemical compounds reveal a number of substances from which a biological activity has already been reported. Here you should include two additional references from Saeidnia et al. (2012) and Jeliani et al. (2018). I have added the full reference in the commented manuscript.

Authors’ response: References added (P11L313 & P15L536) and (P7L240 & P14L528).

  • The 2nd part is on fungicidal/fungistatic effects IN VITRO. You should emphasize this in your manuscript that you test only in vitro, but not in-vivo. So the effect in a real plant-pathogen interaction still needs to be determined. But to my knowledge it is the first time that extracts from Gracilariopsis persica have been used against these fungi.

Authors’ response: Corrected (P1L1).

Critical evaluation

  • I found the study on the compounds sound and I could not find any points of critics here.

2)   The fungicidal test has two critical points: (a) the solvent of the extracts remain unclear to me. The extract was made with methanol and I could not find information, if the solvent was removed and exchanged by water. If methanol was the solvent (which I assume from your manuscript) a solvent control is missing. It is known that methanol could also alter fungal growth and it is scientific standard to use a solvent control for checking this. (b) I find the idea to check for fungistatic effects good, but in your results you just report very briefly on the outcome of this experiment. You have to add some data here. Otherwise, you should not term the effect antifungal BUT fungistatic only, because you could not show an fnugicidal (antifungal) effect clearly. The manuscript should be clearer on this point.

Authors’ response: In the materials and methods, the method of extraction, which has been done with methanol, is stated. After extraction, the methanolic extract was dried by evaporator and finally during application, dissolved in distilled water, which was added to the materials and methods in one line (P3L139-140).

Regarding the fungicidal effect, as stated in the conclusion and results, this extract had more Inhibitory effects on fungi and did not show a fungicidal effect (P10&11L286-306).

3) L81. Here information on algae are missing. There is an abrupt switch from fungicides to "these compounds", which refer to compounds from algae. Please add a general paragraph or sentence on algae and the use of their compounds to bridge from fungicides to algae.

Authors’ response:  Added (P10&11L286-306).

Reviewer 2 pdf File

1) Add in-vitro in the title

Authors response :  In-vitro was added to title   

  • May be could add here a sentence that moisture on plants is a mian factor for fungal infections of these four fungal diseases.

Authors’ response: Added to text.

  • I suggest to introduce here the term "biorationals" which is more and more used for these compounds.

Authors’ response: The referee's suggestion was applied (P2L82-83).

  • what was the solvent? methanol? I miss a blank control of the solvent.

Authors’ response:  Necessary explanations were added to the Materials and Methods section (P2L139-140).

  • Fungicidal: when NO mycelial growth could be observed
  • Fungistatic: when mycelial growth could be observed of the fungus which showed no growth in the 1st experiment. In this case it would be shown that the fungus is still alive and re-growth is possible.
  • Could please revise center to make clear.

Authors’ response: Corrected (P5L207 & 215-216).

Please ad in all graphs and table standard deviation (SD) of the mean and not the standard error (SE).

Please add information on numbers of replicates (n) for each results.

Please add information, if you have tested for normal distribution and variance homogeneity.

Authors’ response:

SD was added instead of SE in all graphs (P9 & P10)

Number of replications were added (P5L221).

For all data, before analysis, Kamogrov-Sminov and Shapiro-Wilk tests were performed for normal distribution and homogeneity variance.

could please check the given names. I am not sure in any case if the notation is following the international nomenclature. Thank you.

Authors’ response: The names were checked. The mentioned names are another name of these compounds, which are also mentioned in highlighted form as follow:

Authors’ response: 3.447; Boron, trihydro (N-methylmethanamine-, (T-4)-:

Formula: C2H9BN; Other names or synonyms: Boron, trihydro(N-methylmethanamine)-, (T-4)-; Dimethylamine, compd.

4.912;  2(3H)-Furanone, dihydro-2(3H)-Furanone, dihydro-4-hydroxy-

Molecular Formula

C4H6O3

;6.649;  2,3-O-(ethoxymethylene)-:

 (+)-2,3-O-Ethoxymethylene-d-ribonolactone

12.900;            5-Thiazoleethanol, 4-methyl

Molecular Formula: C6H9NOS; Other names or synonyms: 4-Methyl-5-Thiazoleethanol

17.380;            2-(1,4,4-Trimethyl-cyclohex-2-enyl)-ethanol:

Molecular FormulaC11H20;

18.170;            10-Methyl-9-oxabicyclo[6.4.0]dodecan-1(8)-ene

Molecular Formula     C12H20O

Synonyms: 10-Methyl-9-oxabicyclo[6.4.0]dodecan-1(8)-ene

110288-22-7

10-Methyl-9-oxabicyclo(6.4.0)dodecan-1(8)-en

2-Methyl-3,4,5,6,7,8,9,10-octahydro-2H-cycloocta[b]pyran

20.213;            Methanone, diphenyl-

Molecular Formula     C13H10O or C6H5COC6H5

synonyms: BENZOPHENONE, diphenyl methanone, Diphenyl ketone; Methanone, diphenyl-

20.831;            Cyclododecasiloxane, tetracosamethyl-

Molecular Formula     C24H72O12Si12

Synonms: Cyclododecasiloxane, tetracosamethyl; Tetracosamethylcyclododecasiloxane; Cyclododecasiloxane, tetracosamethyl.

24.230;            Cyclododecasiloxane, Tetracosamethyl-

Molecular Formula     C24H72O12Si12

Synonms: Cyclododecasiloxane, tetracosamethyl; Tetracosamethylcyclododecasiloxane; Cyclododecasiloxane, tetracosamethyl.

25.951;            2-Propenoic acid, 2-methyl-

Molecular Formula: C4H6O2

Synonyms: Methacrylic acid; α-Methylacrylic acid; Methylacrylic acid; 2-Methyl-2-propenoic acid; 2-Methylacrylic acid; CH2=C(CH3)COOH; 2-Methylpropenoic acid; Methacrylic acid glacial; α-Methacrylic acid; Acrylic acid, 2-methyl-; Propionic acid, 2-methyl-; Kyselina methakrylova; NSC 7393.

26.942;            1-Decanol, 2-hexyl-

Molecular Formula: C16H34O

Synonyms: 2-Hexyl-1-decanol; 2-Hexyldecan-1-ol; 1-Decanol, 2-hexyl-; 2-Hexyldecanol

45.258;            Cholest-5-en-3-ol (3.beta.)-

Chemical Name: cholesterol;

Synonyms: (3beta,14beta,17alpha)-cholest-5-en-3-ol; cholest-5-en-3beta-ol; Cholest-5-en-3beta-ol; Cholesterin; cholesterol; Cholesterol; CHOLESTERO

Please check quality of photos. Some of them appear distorted. Control plate of P. expansum seems not a single ring.

The photo quality corrected and transfer to supplementary.

With your results from figure 5, you do not need the photos here. There are no obvious optical changes in the fungus or morphology.

Figures were transferred to supplementary

You need to add some data on re-growth of mycelium on fresh medium. Was it different among the fungi. Do they recover completely? Was growth speed comparable to a non-treated fungus.

Please add more detailed information here. Compare the growth to fungus which has not been cultivated on agar containing extract.

Authors’ response: Added detailed information on the result section (P10 & 11L286-306).

Reviewer 3 Report

Dear authors,

you present a study on extract of Gracilariopsis persica. In the first part you characterize the chemical compounds in the extract and in the 2nd part you test the fungicidal / fungistatic efficiacies of the extracts against four major fungal pathogens in vitro.

The first part on chemical compounds reveal a number of substances from which a biological activity has already been reported. Here you should include two additional references from Saeidnia et al. (2012) and Jeliani et al. (2018). I have added the full reference in the commented manuscript.

The 2nd part is on fungicidal / fungistatic effects IN VITRO. You should emphasize this in your manuscript that you test only in vitro, but not invivo. So the effect in a real plant-pathogen interaction still needs to be determined. But to my knowledge it is the first time that extracts from Gracilariopsis persica have been used against these fungi.

Critical evaluation:

(1) I found the study on the compounds sound and I could not find any points of critics here.

(2) The fungicidal test has two critical points: (a) the solvent of the extracts remain unclear to me. The extract was made with methanol and I could not find information, if the solvent was removed and exchanged by water. If methanol was the solvent (which I assume from your manuscript) a solvent control is missing. It is known that methanol could also alter fungal growth and it is scientific standard to use a solvent control for checking this. (b) I find the idea to check for fungistatic effects good, but in your results you just report very briefly on the outcome of this experiment. You have to add some data here. Otherwise, you should not term the effect antifungal BUT fungistatic only, because you could not show an nugicidal (antifungal) effect clearly. The manuscript should be clearer on this point.

L81. Here information on algae are missing. There is an abrupt switch from fungicides to "these compounds", which refer to compounds from algae. Please add a general paragraph or sentence on algae and the use of their compounds to bridge from fungicides to algae.

Author Response

Reviewer 3

  • Title: I suggest  ‘Antifungal activity of the extract of a macroalga, Gracilariopsis persica, against four plant pathogenic fungi’ 

Authors’ response: Corrected (P1L1).

  • Line 42: ‘have long been used to source food,..’ change in ‘have long been used as source of..’ 

Authors’ response: Corrected (P2L48)..

  • Line 44’.. and carrageenan, for industrial products.’ Change in ‘ .. and carrageenan used in many industrial sectors.’ 

Authors’ response: Corrected (P2L50)..

  • Vedere ‘contain in linea 45 e l’have riga 47 

Authors’ response: Corrected (P2L53-56).

  • Line 46-48: Owing to their diversity, algae have various compounds in different growth stages and are regarded as 47 rich sources of compounds generally known as secondary metabolites. These secondary metabolites can display diversity of their secondary metabolites, seaweeds are a potential source of antimicrobial 49 materials and exhibit antiviral, antibacterial, and antifungal activities [5]. 

Authors’ response: Corrected (P2L53-56).

  • I suggest: Depending to their phylum, growth stage and environmental conditions, algae can contain different amount of bioactive compounds, including secondary metabolites that could exhibit antiviral, antibacterial, and antifungal activities (references). 

Authors’ response: Corrected (P2L53-56).

  • Line 50: the authors mentioned a list of activity displayed by algae, but the reference is about only antibacterial activity. I suggest some articles for the other ones to be included: 

Righini, H., Roberti, R. & Baraldi, E. Use of algae in strawberry management. J Appl Phycol 30, 3551–3564 (2018).   

Angélica Ribeiro Soares et al. 2012. Antiviral activity of extracts from Brazilian seaweeds against herpes simplex virus. Rev. Bras. Farmacogn. Braz. J. Pharmacogn. 

Authors’ response: Added as a referenc (P2 L56 & P13 L434-435).

Line 53: ‘are two globally distributed fungi with’ change in ‘..are two globally distributed plant pathogenic fungi with..

Authors’ response:  Corrected.

  • Line 51-55: My suggestion: Plant diseases, especially those caused by plant pathogenic fungi, are key factors in the production and quality of agricultural crops and in postharvest life of fresh fruit and vegetables. Among these pathogens, Pyricularia oryzae ( oryzae) and Botrytis cinerea (B. cinerea) are two globally distributed fungi with a wide range of hosts, ranked as the first and second most important plant pathogen in the world based on their economic and scientific importance [7]. 

Authors’ response: Corrected (P2L57-63)..

  • Line 55-59: Please rewrite. 

Authors’ response: Line 55-59 re-written (P2L62-66).

  • Lines 71-86: Please, rewrite. It is confusing. The authors have mixed many concepts. Follow a logical thread, starting explaining as disease control is usually based on, the problems risen from the use of synthetic products 

Authors’ response: Corrected (P2L82-88).

Round 2

Reviewer 1 Report

The manuscript of Pourakbar et al. has been revised according to the review. There are still some minor issues, which should be clarified:

Introduction

P2: The sentence about microalgae derived compounds is still not correct (there are unnecessary repetitions). It should be something like this: „The amount of compounds derived from various families of macroalgae, including green algae, brown algae, and red algae is estimated at 40,000 [20].

Materials and Methods

P2: The circumstances of drying are still not clear. What does it mean “shady state”?

P3: The term filtrated is still not correct, it should be “filtered”.

P3: What was the volume of the Erlenmeyer flask (00 ml is meaningless…).

P4: It is still not clear, how the control cultures were prepared. The concerning question was answered, but I still not find this sentence in the revised manuscript: “In the control, instead of the extract, the same amount of distilled water was added to the culture medium.”

Results

P8, Fig.3: The chromatograms are still not good representations of phenolic compounds, the peaks are barely visible. As I recommended, the part of the figures containing the peaks should be enlarged, to prove that real peaks are there.

Author Response

Reviewer 1 Report Round 2

The authors are thankful to the reviewers for the excellent Reviewing of the MSS. The suggestion helped in the significant improvement of the MSS. All the suggestions of the reviewer have been incorporated in the MSS.

P2: The sentence about microalgae-derived compounds is still not correct (there are unnecessary repetitions). It should be something like this: „The amount of compounds derived from various families of macroalgae, including green algae, brown algae, and red algae is estimated at 40,000 [20].

Authors Response: Revised as suggested Line 99-102

Materials and Methods

P2: The circumstances of drying are still not clear. What does it mean “shady state”?

Authors Response: corrected as ambient shade drying conditions at the average temperature of 27 °C. Line No. 131

P3: The term filtrated is still not correct, it should be “filtered”.

Authors Response: Corrected as filtered. Line No. 135 and 142

P3: What was the volume of the Erlenmeyer flask (00 ml is meaningless…).

Authors Response: Corrected as 500 ml. Line 134

P4: It is still not clear, how the control cultures were prepared. The concerning question was answered, but I still not find this sentence in the revised manuscript: “In the control, instead of the extract, the same amount of distilled water was added to the culture medium.”

Authors Response: Added in Line 192

Results

P8, Fig.3: The chromatograms are still not good representations of phenolic compounds, the peaks are barely visible. As I recommended, the part of the figures containing the peaks should be enlarged, to prove that real peaks are there.

Authors Response: Corrected page 9.

The Discussion section is not well constructed. There is information about everything required to explain the results, but somehow the text is not coherent as a whole. It definitely should be improved.

Authors Response: Improved.

Reviewer 3 Report

Dear authors,

thank you for submitting the revised manuscript. You have responded to all comments made and I found the manuscript has been improved. I only found some little mistakes, which could be addressed in the editorial process.

I only suggest another proof-reading of the manuscript. I did not comment in the manuscript but grammar, wording, and sentence order should be checked by a native speaker or by a language service.

Otherwise, the manuscript is ready for publication

Author Response

Reviewer 3 Round 2 Report

Thank you for submitting the revised manuscript. You have responded to all comments made and I found the manuscript has been improved. I only found some little mistakes, which could be addressed in the editorial process.

I only suggest another proof-reading of the manuscript. I did not comment in the manuscript but grammar, wording, and sentence order should be checked by a native speaker or by a language service.

Otherwise, the manuscript is ready for publication

Authors response: The authors are thankful to the reviewers for such an excellent evaluation. All the comments as suggested by the referee have been incorporated in the paper and highlighted in green background
